# Beyond Aortic Stenosis: Addressing the Challenges of Multivalvular Disease Assessment

**DOI:** 10.3390/diagnostics13122102

**Published:** 2023-06-17

**Authors:** Sara Bombace, Maria Chiara Meucci, Federico Fortuni, Federica Ilardi, Rachele Manzo, Grazia Canciello, Giovanni Esposito, Paul A. Grayburn, Maria Angela Losi, Anna Sannino

**Affiliations:** 1Leipzig Heart Science, 04289 Leipzig, Germany; 2Department of Cardiovascular Medicine, Fondazione Policlinico Universitario A. Gemelli, IRCCS, 00168 Rome, Italy; 3Department of Cardiology, Leiden University Medical Centre, 2333 ZA Leiden, The Netherlands; 4Department of Cardiology, San Giovanni Battista Hospital, 06034 Foligno, Italy; 5Department of Advanced Biomedical Sciences, Division of Cardiology, Federico II University, 80131 Naples, Italy; 6Baylor Scott & White Research Institute, Plano, TX 75093, USA

**Keywords:** aortic stenosis, multiple valve disease, mixed valve disease, echocardiography, multimodal imaging

## Abstract

Aortic stenosis (AS) can often coexist with other valvular diseases or be combined with aortic regurgitation (AR), leading to unique pathophysiological conditions. The combination of affected valves can vary widely, resulting in a lack of standardized diagnostic or therapeutic approaches. Echocardiography is crucial in assessing patients with valvular heart disease (VHD), but careful consideration of the hemodynamic interactions between combined valvular defects is necessary. This is important as it may affect the reliability of commonly used echocardiographic parameters, making the diagnosis challenging. Therefore, a multimodality imaging approach, including computed tomography or cardiac magnetic resonance, is often not just beneficial but crucial. It represents the future of diagnostics in this intricate field due to its unprecedented capacity to quantify and comprehend valvular pathology. The absence of definitive data and guidelines for the therapeutic management of AS in the context of multiple valve lesions makes this condition particularly challenging. As a result, an individualized, case-by-case approach is necessary, guided primarily by the recommendations for the predominant valve lesion. This review aims to summarize the pathophysiology of AS in the context of multiple and mixed valve disease, with a focus on the hemodynamic implications, diagnostic challenges, and therapeutic options.

## 1. Introduction

Multiple and mixed valvular heart diseases (VHD) are highly prevalent conditions, expected to increase in prevalence as the population gets older [1]. The clinical pattern and the combination of the involved valves are extremely variable, thus leading to the absence of a standardized diagnostic or therapeutic approach in this setting. Since aortic stenosis (AS) is the most frequent VHD worldwide, it represents one of the most frequent findings in the context of multiple VHDs.

The diagnostic algorithm should always start with the identification of the underlying etiology as well as the concomitant or VHD-induced cardiac structural changes. Moreover, in the presence of combined valvular defects, it is of pivotal importance to be aware of their hemodynamic interaction since this could affect the reliability of most of the echocardiographic methods, making the diagnosis very challenging. For this reason, in these complex scenarios, a multimodality approach using different imaging techniques, such as computed tomography (CT) or cardiac magnetic resonance (CMR), often represents the keystone of the management strategy. Apart from the definition of the appropriate diagnostic modalities, a case-by-case, individualized therapeutic management should always be chosen, mainly driven by the guideline’s recommendations for the predominant valve lesion [1,2].

The absence of clear data and, thus, guidelines for the management of AS in the context of multiple valve lesions makes this condition a great challenge for the clinician. In this review, we will discuss the pathophysiology of AS in the context of multiple valve disease, its diagnostic challenges, and therapeutic options.

## 2. Pathophysiology of Concomitant Aortic Stenosis and Other Valve Lesions

AS and mitral regurgitation (MR) are the most common VHDs in developed countries [3]. The high prevalence of these conditions increases the likelihood of observing their co-occurrence, although the pathophysiology of multiple valve disease is often complex and multifactorial.

Multiple valve diseases can arise from a variety of etiologies, including functional valve disease, rheumatic heart disease (RHD), degenerative processes, infiltrative cardiomyopathies, congenital heart diseases (CHD), and infectious endocarditis (IE), among others. Understanding the underlying cause of multiple valve disease is crucial for accurate diagnosis, management, and treatment of this complex scenario.

### 2.1. Functional Valve Disease

Functional valve disease commonly co-occurs with AS and can arise from various mechanisms. Functional MR is observed in 63% of patients with AS [4], whereas tricuspid regurgitation (TR) is in 40% [5]. The pressure overload imposed by severe AS can lead to decreased left ventricular (LV) function and eccentric LV remodeling. Additionally, patients with AS are at higher risk for coronary artery disease and ischemic cardiomyopathy [6], both of which may exacerbate LV dilatation and dysfunction, cause leaflet tethering, and finally result in secondary MR [7]. AS can also cause diastolic dysfunction and increase left atrial pressure, leading to backward pressure in the pulmonary veins and the development of pulmonary hypertension (PH) [8]. The elevated pressure in the pulmonary circulation increases right ventricular (RV) afterload, resulting in right atrial (RA) and RV dilation, leaflet tethering and malcoaptation, and consequent TR [9]. Furthermore, prolonged increases in left atrial pressure due to chronic afterload can cause atrial dilatation, promoting atrial fibrillation, which is also associated with functional TR and MR (Figure 1) [7].

### 2.2. Rheumatic Heart Disease

RHD is a common cause of multiple valve disease, particularly in developing countries [10], but has become a global illness due to ongoing immigration and emigration [11]. RHD is the long-term complication of acute rheumatic fever (ARF) due to immune-mediated cardiac injury caused by group A β-hemolytic streptococci (also known as S. pyogenes) infection [12,13]. The cumulative incidence of RHD at 10 years after AFR is 51.9% [14].

Valvular thickening, restricted leaflet mobility, and nodularity along the length of the leaflet are typical morphological features [15]. The mitral valve (MV) is the most affected valve, followed by the aortic valve (AV), while the tricuspid and pulmonary valves are rarely involved [16]. Rheumatic AV involvement usually occurs in the presence of rheumatic MV disease, and regurgitation is more common (47%) than stenosis (14%) [17]. In addition, rheumatic AS usually coexists with some degree of aortic regurgitation (AR) due to the retraction of cusp edges [17].

### 2.3. Degenerative Etiology

Degenerative etiology is a prevalent cause of multiple valve disease, particularly in developed countries where aging is a significant contributing factor. It is estimated that approximately 10% of people over the age of 75 are affected by some form of VHD [18].

The degenerative process in valve tissue is characterized by progressive calcification and thickening of valve leaflets. These changes result in increased stiffness of the valve tissue, leading to stenosis or regurgitation. The most affected valves are the AV and MV. Calcific AV disease is characterized by progressive thickening and calcification of the AV without commissural fusion, while degenerative MV disease involves progressive calcification of the mitral annulus at the base of the leaflets or degenerative changes of the MV causing prolapse [19].

The presence of comorbidities such as hypertension, metabolic syndrome, or chronic kidney disease can also contribute to valve degenerative processes. These conditions cause mechanical stress on the valves, alterations of the metabolism with toxin accumulations, increased oxidative stress, and inflammation, ultimately accelerating valve dysfunction [20,21]. It is important to note that these chronic conditions often coexist in the same individual and increase with age, and the presence of concomitant risk factors can further increase the risk of valve disease.

### 2.4. Infiltrative Cardiomyopathies

Infiltrative cardiomyopathies, including amyloidosis and Fabry disease, can also affect heart valves, leading to valve disease [22,23,24]. In particular, the coexistence of AS and cardiac amyloidosis (CA) is not uncommon in the elderly [25].

In CA, the extracellular deposition of amyloid fibrils can lead to thickening and stiffening of the heart valves. The infiltration of amyloid substances in the AV can result in stenosis, whereas infiltration in the MV typically leads to regurgitation [22]. Similarly, in Fabry disease, the accumulation of glycosphingolipids in valve fibroblasts may lead to valve disease, more commonly affecting left-sided valves [23].

### 2.5. Congenital Heart Disease

AS can be present alongside CHD, which can also impact other valves [26]. The most common cause of congenital valvular AS is bicuspid AV (BAV), which is found in up to 2% of the population [26]. A BAV only has two cusps instead of the normal three, which can result in early or accelerated AV dysfunction, including stenosis or regurgitation; mixed valvular disease is also possible. Additionally, individuals with BAV have an increased risk of developing multiple valve diseases. Recent studies have shown that BAV is a complex genetic disorder that can be isolated or associated with other genetic syndromes. However, even in patients without syndromic features, BAV is associated with congenital heart and vascular abnormalities, such as coarctation of the aorta (7%), patent ductus arteriosus (8.5%), MV abnormalities (11%), ventricular septal defects (14%), and thoracic aortic aneurysm (50%) [27,28,29].

### 2.6. Infective Endocarditis

IE is a condition where the valvular endocardium is infected, with the AV being the most affected valve [30]. The causative organisms, acute inflammation, and thrombi contribute to the formation of vegetation on the valve cusps. This process can lead to AS, worsening of the degree of a pre-existing AS, or destruction of the valve tissue that may result in regurgitation. It is also possible for both conditions to coexist. On the other hand, IE is a well-known complication of AS, and its incidence increases with the severity of AS. Turbulent flow through a narrowed AV can damage the endothelial lining, leading to the deposition of platelets and fibrin, which can act as a nidus for bacterial colonization and subsequent infection. The risk of developing IE is further increased in patients with pre-existing valve calcification or regurgitation, such as rheumatic disease or congenital BAV [31]. Due to the fibrous continuity between the AV and the anterior leaflet of the MV, AV endocarditis can easily spread to involve the MV [32].

### 2.7. Other Causes

Less frequently, severe AV may be a long-term sequelae of mediastinal radiation therapy. Radiation damage affects more commonly left-sided heart valves, leading to their thickening and calcification, and it usually occurs after a latency period of several years [33].

## 3. Diagnostic Challenges

The coexistence of multiple VHDs represents one of the most challenging cardiovascular conditions for a cardiovascular imaging specialist [1]. Indeed, the majority of echocardiographic parameters for VHD quantification have been validated in patients with only one affected valve and may be misleading in the presence of multiple VHDs [1]. Moreover, the impact of a valve disease on chamber size and performance, as well as flow status, may be modulated by the presence of other concomitant valvular lesions. Thus, an integrative approach using different imaging modalities becomes of crucial importance for a proper diagnosis in patients with multiple VHDs, with the AS being the most common valvular defect involved [1].

### 3.1. Role of Echocardiography

Aortic Stenosis with concomitant Mitral Regurgitation. In patients with concomitant AS and MR, a quantitative approach to the estimation of MR severity is encouraged. As systolic intraventricular pressure is increased due to AS, concomitant MR is characterized by increased transmitral systolic velocity that results in a disproportional increase in color-flow jet area and regurgitant volume [34]. Therefore, the assessment of MR should be based on the vena contracta and effective regurgitant orifice area (EROA) calculations with the proximal isovelocity area method (PISA), which are not afterload-dependent and thus more reliable and representative of the true MR severity. Moreover, in patients with MR undergoing AV replacement, as the reduction in LV systolic pressure contributes to the decrease in regurgitant volume, EROA proved to be a true marker of lesion severity [35]. For what concerns AS evaluation, it should be noted that moderate or severe MR might cause an underestimation of the transaortic pressure gradient as an effect of the reduction on the net forward flow across the AV, which is therefore associated with the development of a low-flow, low-gradient (LF-LG) AS. This LF-LG AS pattern may occur with a reduced (classic LF-LG AS) or preserved LV ejection fraction (paradoxical LF-LG AS). In this circumstance, a functional AV area (AVA) calculation might be helpful. Alternatively, an integrative approach, including AV calcium quantification by multi-detector CT (MDCT), should be preferred. Moreover, distinguishing a high-velocity MR jet from the AS jet can present a challenge as both are systolic signals directed away from the apex. However, the timing of the signal and the shape of the velocity curve can aid in differential diagnosis. Specifically, the MR jet is longer in duration and characterized by an early peak that quickly decays, whereas the AS jet typically has a more rounded peak that occurs later in systole (36)

Aortic Stenosis with concomitant Mitral Stenosis. Similar to MR, severe mitral stenosis (MS) associated with AS leads to a great reduction in cardiac output, resulting in low flow rates and pressure gradients, with a possible underestimation of the severity of both valves if only Doppler measurements are considered [36]. Paradoxical LF-LG is commonly observed and might require AV calcium quantification by MDCT. The continuity equation for the calculation of the AVA and MV area is not reliable because of its dependency on flow conditions. In addition, the pressure half-time (PHT) method, which depends on the pressure difference between two chambers, should not be used since the altered compliance of the LV due to the presence of AS might overestimate the MV area [37]. In these circumstances, anatomical measurement of the MV area is considered the most reliable method, eventually using real-time 3D transesophageal echocardiography to provide a better alignment of the image plane at the mitral tips and thus a more accurate MV orifice area definition [38]. When technically feasible, the PISA method offers an option for the quantification of the MV area (Figure 2).

Aortic Stenosis with concomitant Aortic Regurgitation. The concomitant presence of significant AS and AR poses several diagnostic challenges. The impaired LV relaxation secondary to the pressure overload makes the PHT method unreliable for the evaluation of AR [39]. Conversely, the EROA or regurgitant volume reflects the severity of AR, even if it must be noted that, in patients with mixed AV disease, the LV volume is smaller than in those with pure AR, thus making the regurgitant fraction higher [40]. The increased stroke volume related to the AR is responsible for increased aortic outflow velocity and pressure gradients that might overestimate AS, but these parameters may be useful to assess the overall hemodynamic severity of AV disease [2]. In addition, the simplified Bernoulli equation for calculation of the AV pressure gradient should not be used due to the increased LV outflow tract velocities.

In mixed AV disease, the assessment of AVA remains accurate using the continuity equation; however, AVA may increase at high transvalvular flow rates, and, in some patients, an AVA greater than 1.0 cm^2^ might reflect severe AS. Doppler velocity index (ratio of the velocity time integrals in the LV outflow tract versus the aortic jet) is not significantly affected by the presence of AR and, together with the effective AVA by the continuity equation method, represents the best parameters to grade AS severity in the context of mixed AV disease. The anatomic AVA measured by planimetry may help corroborate AS severity. Transesophageal echocardiography, with its superior image resolution compared to transthoracic echocardiography, offers an effective modality for direct planimetry of the AVA. This technique involves tracing the contours of the valve orifice during mid-systole, thereby providing a direct measure of the AVA. MDCT or CMR imaging can also accurately measure the anatomic AVA through planimetry. The choice of imaging modality depends on the specific clinical situation, availability of resources, and expertise [41,42].

Aortic Stenosis with concomitant Tricuspid Regurgitation. In the physiopathology of the AS, functional TR is present in one third of the patients, necessitating careful echocardiographic evaluation. While the evaluation of TR severity is not influenced by the coexistence of AS, when TR is chronic and severe, a low-flow condition may superimpose, making the classical continuity equation not reliable in the estimation of AS severity, with a tendency to be significantly underestimated [43,44]. Additionally, an invasive evaluation of cardiac output with the thermodilution method may underestimate the calculated AVA by the Gorlin equation and overestimate AS severity [45].

### 3.2. The Role of Stress Echocardiography

As previously described, the combination of AS with MR or MS is associated with a LF-LG state that makes the estimation of AS severity challenging. Low-dose (≤20 μg/kg per min) dobutamine stress echocardiography (DSE) may be helpful to distinguish true severe from pseudo-severe AS and to assess LV flow reserve when the pressure gradient is low and the LV ejection fraction is reduced. However, in the presence of significant MR, DSE may fail to induce a significant increase in LV outflow and may thus not allow the confirmation of AS severity.

Again, in patients with concomitant AS and AR with reduced LV ejection fraction, a low-dose DSE may be helpful to assess the presence of LV contractile/flow reserve and confirm the overall hemodynamic severity of mixed AV disease. It has been suggested that an increase in peak jet velocity ≥4 m/s and/or a mean gradient ≥40 mm Hg with dobutamine stress could be indicative of severe AV disease [2].

Currently, there is limited data on the use of stress echocardiography in patients with multiple and mixed VHD. Exercise echocardiography, performed using either a treadmill or bicycle ergometer, may be useful to discriminate symptoms in apparent non-critical valve disease at rest that worsen during stress, producing a disproportionate increase in the transvalvular pressure gradient or pulmonary arterial pressure [46]. On the other hand, stress testing can be used to unmask symptoms, abnormal blood pressure responses, or signs of ischemia in those who are apparently asymptomatic despite resting hemodynamic signs of severe AS [47]. An increase in transvalvular gradients, elevations in pulmonary artery pressure, and the absence of LV contractile reserve provide incremental prognostic value over resting echocardiographic parameters that might affect the appropriateness and timing of intervention [48].

### 3.3. Role of CT

In the latest decade, cardiac CT has emerged as a valuable, complementary technique in the evaluation of AS severity through the calculation of the AV calcium score. The latter enables an accurate assessment of the burden of AV calcification and has been validated as a marker of AS severity [49]. The AV calcium score should be calculated on non-contrast electrocardiogram-gated CT scans using the Agatston method [49]. When evaluated with this approach in patients with AS, a sex-specific threshold of 1300 AU for women and 2000 AU for men is strongly indicative of severe stenosis [49]. Considering this evidence, international guidelines [50] advocate the use of CT-derived calcium scoring in the assessment of AS severity when echocardiographic data are inconclusive, whose emblematic example is the LF-LG AS. With respect to most of the echocardiographic parameters for AS grading, the AV calcium scoring has the important advantage of being independent from the hemodynamic status. This strength becomes crucial in the scenario of low-flow conditions, as commonly observed in the presence of multiple VHDs. Indeed, the coexistence of a significant MR or stenosis as well as TR typically leads to a reduction of the forward flow and, thus, is often associated with a pattern of LF-LG AS.

Particularly, the latest international recommendations suggest the use of AV calcium scoring in patients with LF-LG AS in the two following scenarios: (1) LF-LG AS with preserved LV ejection fraction (paradoxical LF-LG AS); and (2) LF-LG AS with reduced LV ejection fraction (classic LF-LG AS), when there is a negative flow response on stress echocardiography.

Despite not yet being widely used in daily clinical practice, AV calcium scoring may represent a faster and potentially easier tool than stress echocardiography, being both feasible and conclusive in many patients [49,51,52]. Moreover, given the essential role of cardiac CT in the pre-procedural planning of transcatheter AV replacement (TAVR), it may be easily obtained by acquiring non-contrast images in addition to the TAVR imaging protocol.

Of importance, aside from its diagnostic value, the CT-derived AV calcium scoring has also been shown to be a strong prognostic marker as a predictor of disease progression and survival, independently from clinical and Doppler echocardiographic data [53,54].

The measurement of the anatomic AVA using planimetry on contrast-enhanced scans represents an alternative method for AS grading by cardiac CT. Although this approach demonstrated a good correlation with echocardiographic data obtained by the continuity equation, it is associated with a systematic overestimation of the AVA and thus should be used with caution [55].

Finally, for patients being considered for surgery, CT offers precise measurement of the thoracic aorta and the capability to exclude coronary artery disease [41].

To this end, CT represents an important, complementary tool in refining AS grading and may provide crucial diagnostic and prognostic information, especially in the setting of low-flow conditions and concomitant VHDs.

### 3.4. Role of CMR

Nowadays, there is limited evidence regarding the specific additive value of CMR in patients with AS and concomitant VHDs. However, CMR appears to be a promising tool in the setting of multiple VHDs since (1) it enables an accurate grading of VHDs, especially of regurgitant lesions, thus overcoming the well-known limitations of echocardiography in these patients; (2) it is the gold standard for the calculation of ventricular dimensions and systolic function; and (3) it has the key strength of myocardial tissue characterization [1].

Regarding the grading of regurgitant lesions in patients with multiple VHDs, phase-contrast CMR imaging with flow quantification in the aorta or pulmonary artery should be the preferred method for calculating the regurgitant volume and regurgitant fraction [56]. Conversely, the quantification of regurgitant volume by measuring ventricular volumes in cine sequences assumes the presence of only one regurgitation lesion and, thus, may be misleading in the context of multiple VHDs [56].

In patients with AS, peak velocity and mean pressure gradient may be derived by phase-contrast sequences, but they are often underestimated in comparison to Doppler data due to the partial volume averaging within the vena contracta [56]. Additionally, CMR enables the quantification of either anatomic or functional AVA, but the application of these measurements in current clinical practice is limited and should be reserved in cases of discordant or incongruent findings with other imaging modalities. Particularly, anatomic AVA may be calculated using planimetry, thanks to the excellent blood-to-myocardium contrast and high signal-to-noise ratio provided by the steady-state free precession sequences [56]. In this regard, a recent meta-analysis [57] reported a high accuracy of CMR-derived anatomic AVA in comparison to transoesophageal echocardiographic data. Given its load-independent nature, it may be a valuable marker of AS severity in the presence of low-flow states and/or concomitant VHDs. However, similarly to planimetric measurements obtained with other imaging modalities, it is less than an optimal method since the visualization of the true valve orifice may be challenged by jet turbulence, leaflet calcifications, and its complex, three-dimensional shape [57]. Alternatively, functional AVA may be calculated by phase-contrast velocity mapping of the velocity-time integral in the LV outflow tract and AV orifice, but little is known regarding its concordance with other diagnostic approaches [58,59].

More importantly, CMR allows for accurate assessment of serial changes in chamber dimensions and ventricular performance that reflect the overall burden of AS and concomitant valvular lesions [1]. In addition, CMR offers the unique opportunity to non-invasively assess myocardial tissue characterization. Indeed, CMR allows the detection of both replacement myocardial fibrosis by late gadolinium enhancement (LGE) imaging and diffuse interstitial fibrosis by calculating myocardial extracellular volume (ECV) on T1 mapping [60].

In patients with AS, the presence of replacement fibrosis on LGE imaging correlates with more severe valve stenosis and LV remodeling [61] and does not regress after AV replacement [62,63], indicating an advanced and irreversible stage of the disease [61]. On the other hand, CMR parametric mapping with ECV quantification allows the detection of diffuse interstitial fibrosis that precedes the development of LGE and, in recent years, has gained attention as a marker of an early and reversible phase of LV remodeling in patients with AS [60,63]. Moreover, similarly to LGE, ECV has also been shown to be a powerful and independent predictor of mortality in AS patients undergoing AV replacement [64]. Thus, while still an emerging parameter, current evidence suggests that ECV may represent a promising tool to identify the myocardial overload in patients with AS and concomitant VHDs before the occurrence of irreversible damage and, importantly, to refine the optimal timing for intervention.

## 4. Therapeutic Approach

### 4.1. Therapeutic Approach to the Patient with Aortic Stenosis

Decisions about treatment can only be made after conducting a thorough cardiac examination. It is crucial to acknowledge that treatment choices may not always be confined solely to the AV. In certain situations, such as IE or infiltrative diseases, treatment considerations may need to address issues beyond the AV itself. Should this comprehensive evaluation reveal that the patient’s primary condition is aortic stenosis, AV replacement (AVR) represents the only therapeutic option that has shown excellent long-term results in the management of severe AS [65]. Indeed, no medical therapy was shown to be effective either to treat severe AS or prevent AS progression [50,66]. Balloon aortic valvuloplasty can provide a temporary increase in AVA, which can transiently improve symptoms without any survival benefit in adults [67]. Therefore, it is still mentioned in the European guidelines, but it may only be considered as a bridge to AVR in hemodynamically unstable patients or in those who require urgent non-cardiac surgery [50]. AVR can be accomplished via surgical and transcatheter approaches. The choice regarding the approach and device type requires careful consideration of the patient profile, longevity, and specific risk factors related to the patient and the specific procedure, which should be assessed by the heart team and possibly in specific heart valve clinics (Table 1) [50,66]. Procedural risk of morbidity and mortality can also be adjudicated by using specific risk scores (such as the Society of Thoracic Surgeons Predicted Risk of Mortality and the European System for Cardiac Operative Risk Evaluation II), which are regularly updated and based on large registries [68,69]. Surgical AVR (SAVR) can be performed with mechanical, stented, or stentless biological prostheses, homograft tissue, or pulmonic root autograft with simultaneous pulmonic root homograft replacement (Ross procedure), whereas TAVR can be performed using balloon- or self-expandable bioprostheses [65].

SAVR can be performed through traditional medial sternotomy or using minimally invasive approaches that have the advantages of reducing the need for blood transfusion, the length of the intensive care unit stay, hospital costs, and the incidence of procedural complications such as renal failure and post-operative atrial fibrillation [70,71]. Nevertheless, they compromise the ability to address concomitant pathologies such as other concomitant VHD and coronary artery disease; moreover, the management of surgical complications may be more complex. The main advantage of mechanical prostheses is their durability; indeed, modern mechanical prostheses have virtually no degeneration [65]. Their design has improved through the years. Nowadays, bileaflet mechanical valve prostheses are more often used, with advantages in terms of lower gradients, minimal rates of regurgitation, and lower risk of thromboembolism with less intense anticoagulation regimens compared with previous designs. However, long-term systemic anticoagulation with warfarin is still recommended to reduce thromboembolic complications, with a typical international normalized ratio (INR) range of 2.0 to 3.0. The downside is the increased incidence of bleeding events, which is about 1% per year in patients with mechanical prosthetic valves [72]. Ongoing studies are testing the use of lower INR ranges [73] and the use of direct oral anticoagulants [74] in patients with modern-design mechanical valves.

Differently from mechanical valves, bioprostheses can be implanted both surgically and with a transcatheter approach. The main advantage of bioprostheses is that they do not require long-term anticoagulation. Nevertheless, since they are composed of biological material (essentially porcine or bovine pericardium), they undergo degeneration and are therefore less durable compared with mechanical valves [75]. The transcatheter approach is generally performed through the common femoral artery (>95%) [65] and, when not doable, through other peripheral vascular accesses or a transapical approach. Compared to SAVR, TAVR does not require the institution of cardiopulmonary bypass and cardioplegic arrest of the heart, can be performed with local anesthesia with or without conscious sedation, and therefore is preferred in fragile and high-risk patients [50,66]. Compared with SAVR, TAVR in patients with concomitant VHD might be more challenging and sometimes not feasible with the transcatheter approach [76]. Perioperative mortality between SAVR and TAVR has been shown to be comparable [77,78]. Nevertheless, TAVR has been associated with higher rates of permanent pacemaker implantation and paravalvular leaks due to prosthesis malposition compared with SAVR [79]. However, only moderate or severe paravalvular leaks, which are less common compared to the mild ones, showed an association with worse long-term outcomes [80].

### 4.2. Therapeutic Approach to the Patient with Aortic Stenosis and Concomitant Valvular Disease

In cases of concomitant VHD, clinical decision-making can be challenging, and an individualized, case-by-case multidisciplinary approach is necessary. There are essentially 2 possible scenarios: (1) AVR is indicated because of severe AS in the presence of concomitant mitral or tricuspid valve disease; and (2) mitral or tricuspid valve surgery is indicated in the presence of concomitant AS. In these two conditions, the physician should follow the current guidelines [50,66] applicable to the predominant VHD [76]. When both VHDs are severe, the consensus is that they should both be treated [76]. However, the management of less-than-severe concomitant VHD remains controversial. The first step is to determine the predominant valvular lesion. Several general considerations can help guide this process. First, the severity of each valve lesion should be assessed using various imaging modalities, such as echocardiography, CMR, or CT scans. The valve with the most severe stenosis or regurgitation may be the primary problem. Second, the valve with the most significant impact on hemodynamics, such as causing left ventricular dysfunction or pulmonary hypertension, may be the primary problem. Third, if one valve lesion appeared earlier than the others, it may have been the primary problem that led to the development of other valve lesions. Finally, symptoms related to a specific valve lesion may indicate that it is the primary problem.

In clinical practice, two commonly encountered scenarios involve AS accompanied by either MR or TR.

Whether the presence of concomitant moderate MR is associated with worse outcomes after AVR remains a matter of debate [81]. AVR reduces LV systolic pressure and hence also MR driving pressure. Moreover, reverse LV remodeling with regression of hypertrophy and dilatation and improvement of systolic function has been documented [63]. All these changes justify the MR improvement observed in most patients after AVR [82]. Based on these considerations, compared to primary MR, secondary MR and a high probability of LV reverse remodeling are associated with a larger reduction in MR severity after AVR.

Concomitant significant secondary TR has been associated with worse outcomes in patients with severe AS in several observational studies [83]. Nevertheless, TR may be secondary to multiple conditions (such as atrial fibrillation, pulmonary hypertension, and RV dysfunction), which can have a per se negative impact on patient prognosis. Therefore, whether secondary TR is itself independently associated with a worse prognosis or is merely a surrogate marker of disease remains a matter of debate [76]. Moreover, the fact that TR severity is dynamic and varies with loading conditions has led to the conclusion that tricuspid annulus diameter (>21 mm/m^2^), which has been associated with TR worsening [84], rather than TR severity should be considered to decide whether concomitant TR should be treated at the time of AVR. To facilitate the decision-making process for both scenarios, a proposed algorithm (Figure 3 and Figure 4) that takes into account the severity of the regurgitation, morphological and hemodynamic factors, and the individual’s operative risk has been proposed [50,85,86].

## 5. Future Aspects and Gaps in Evidence

Optimal grading for multiple or mixed VHDs still needs to be established. Nonetheless, the integration of multimodality imaging, including techniques such as CT and CMR, holds great promise for the future of diagnostics in this complex field. Despite the inherent challenges and costs associated with the utilization of multiple advanced imaging modalities, there is a justifiable rationale for employing multiple tests, given their unparalleled capabilities in quantifying and comprehending the intricacies of valvular pathology. Future research efforts should focus on validating the utility and effectiveness of these imaging techniques in clinical practice. Additionally, exploring novel imaging modalities, such as 4D flow CMR, particle image velocimetry, and vector flow mapping in echocardiography, could potentially enhance the accuracy of VHD diagnosis and management. Although these techniques are currently confined to research settings, their future application holds substantial potential to enhance patient care [87].

## 6. Conclusions

In conclusion, AS combined with other valvular diseases or AR represents a complex and challenging clinical scenario. Echocardiography is crucial, but careful consideration of hemodynamic interactions between combined valvular defects is necessary. A multimodality imaging approach may also be required. Therapeutic management lacks definitive guidelines, making an individualized, case-by-case approach necessary. Further research and studies are needed to establish more standardized diagnostic and therapeutic approaches for this challenging condition.

## Figures and Tables

**Figure 1 diagnostics-13-02102-f001:**
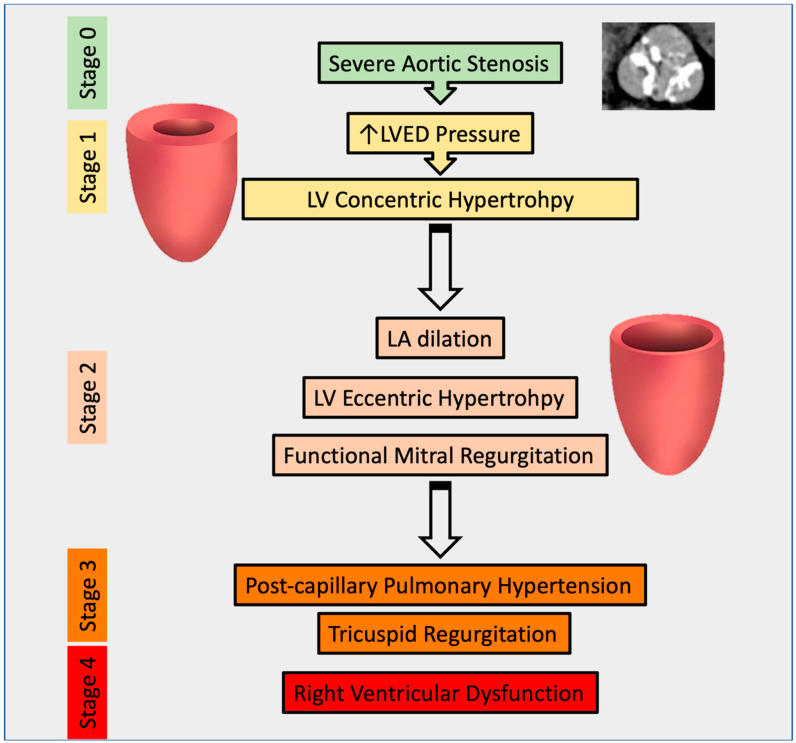
Pathophysiology of Aortic Stenosis by disease stage. In this figure, an illustration of the progression of AS is provided, with particular emphasis on the pathophysiological basis for the occurrence of functional mitral and tricuspid regurgitation.

**Figure 2 diagnostics-13-02102-f002:**
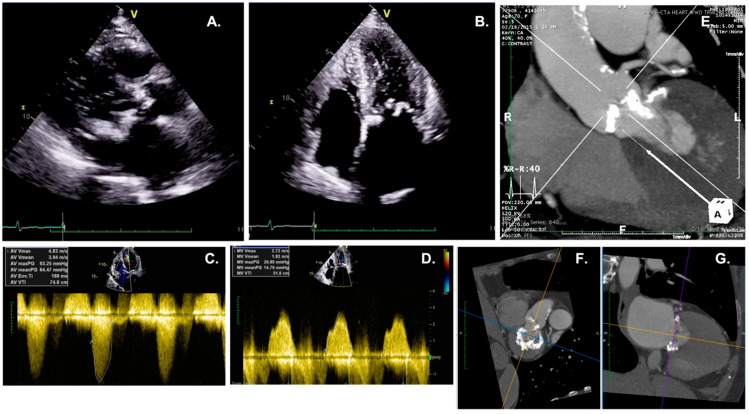
A case of concomitant severe AS and MS due to severe mitral annular calcification. (**A**,**B**). Representative echocardiographic images from parasternal long-axis view and apical 4 chamber. (**C**,**D**). Continuous wave Doppler on the AV and MV, showing high gradients. (**E**–**G**). CT images of this same patient help identify the distribution of calcium within the LVOT, aortic cusps, and mitral annulus. AS: aortic stenosis; MS: mitral stenosis; AV: aortic valve; MV: mitral valve; CT: computed tomography; LVOT: left ventricular outflow tract.

**Figure 3 diagnostics-13-02102-f003:**
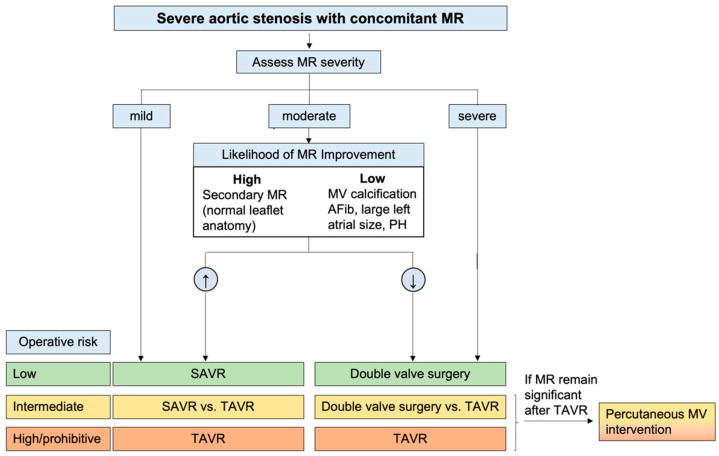
Proposed algorithm for the management of severe aortic stenosis with concomitant mitral regurgitation. MR—mitral regurgitation; MV: mitral valve; AF: atrial fibrillation; PH: pulmonary hypertension; SAVR: surgical aortic valve replacement; TAVR: transcatheter aortic valve replacement. Adapted from Unger 2016 and Kiriyama 2022 [85,86].

**Figure 4 diagnostics-13-02102-f004:**
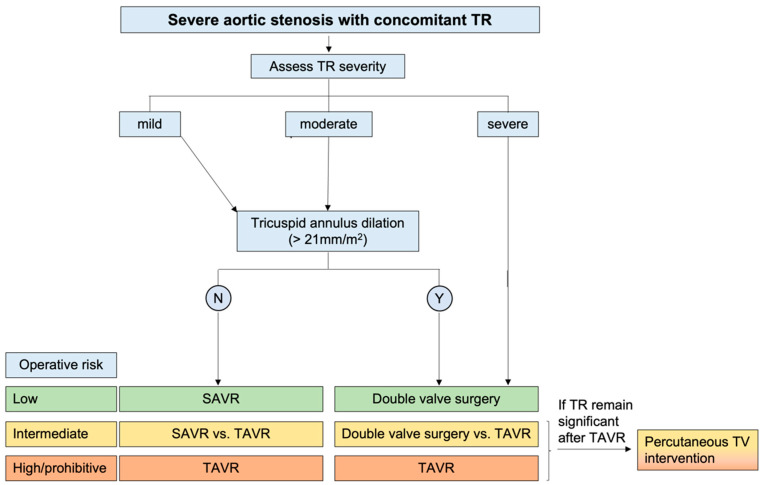
Proposed algorithm for the management of severe aortic stenosis with concomitant tricuspid regurgitation. TR: tricuspid regurgitation; SAVR: surgical aortic valve replacement; TAVR: transcatheter aortic valve replacement; TV: tricuspid valve. Adapted from Vahanian 2022 [50].

**Table 1 diagnostics-13-02102-t001:** Factors favoring SAVR or TAVR.

SAVR	TAVR
Young patients. No controindication to anticoagulation. Prosthesis durability. Low surgical risk. Need to treat concomitant VHD or CAD that requires CABG.	Old patients. High or prohibitive surgical risk. Patient preference. No need to treat concomitant VHD or CAD that requires CABG. Concomitant VHD feasible for trascatheter treatment. Valvular anatomy and vascular access suitable for TAVR.

SAVR: surgical aortic valve replacement; TAVR: transcathter aortic valve replacement; VHD: valvular heart disease; CAD: coronary artery disease; CABG: coronary artery bypass surgery.

## Data Availability

Not applicable.

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
