# Peer review of "Beyond Aortic Stenosis: Addressing the Challenges of Multivalvular Disease Assessment"

_diagnostics, 2023, doi:10.3390/diagnostics13122102_

Round 1
Reviewer 1 Report
This is a review article where the authors aim to elucidate the diagnostic challenges surrounding aortic stenosis. They discuss the various possibilities that need to be considered and provide commentary on different imaging examinations, sharing their professional opinions. The topic is significant, and the English language used is fluent. Below, I will provide my considerations regarding the overall structure and specific details.
1. The authors used the term "dilemma" in the title, but the actual meaning of dilemma is a situation where there are two difficult choices, both with pros and cons, making it extremely challenging to choose between them. The main difficulty lies in the fact that "neither option is ideal" and one has to be chosen. However, after reading the abstract and content, it seems that the authors intended to convey the meaning of "challenge" rather than the concept of "either-or" dilemma. Would you consider reconsidering the title?
If the authors' use of "dilemma" is intended to refer to choosing between SAVR or TAVR, it doesn't seem to align with the context. This is because not all cases of aortic stenosis necessarily require treatment, and even when treatment is necessary, it doesn't always involve these two options.
2. In terms of overall structure, the authors have extensively explored and expanded upon the topic of aortic stenosis. This is indeed reflective of the clinical reality, as aortic stenosis is often not an isolated condition. Even when considering the management of aortic stenosis alone, it is crucial to first exclude other potential underlying causes. Failing to do so may result in worsening of the patient's condition or even increased mortality, highlighting the importance of a comprehensive approach.
However, the current structure of the article seems somewhat divergent. The authors have indeed presented various diseases that are potentially related to aortic stenosis, some being sequelae following aortic stenosis and others being the causes of aortic stenosis. In the end, the authors conclude that each patient's condition should be evaluated individually. However, this organization lacks clear recommendations or standardized guidelines to follow. It appears to primarily provide a wealth of foundational knowledge on cardiac diseases and advises readers to exercise their own judgment.
I would suggest that the authors consider reorganizing the article by categorizing the important scenarios they believe are valuable for readers. Each scenario can be introduced with a case study, particularly emphasizing those cases where misdiagnosis could lead to poor outcomes or those that are often overlooked. This approach would not only showcase your team's expertise in the field but also provide recommendations for addressing the challenging and unstructured clinical dilemmas in a more focused manner.
3. In practical terms, when we encounter patients with aortic stenosis in clinical practice, the usual approach is to perform a comprehensive cardiac ultrasound examination to assess all four cardiac chambers and valve conditions. This evaluation helps determine whether the aortic stenosis is primary or secondary, and provides an understanding of the overall cardiac function. Treatment considerations are then made based on these findings. I believe your team follows a similar approach. In other words, aortic stenosis is not a condition that requires immediate standalone treatment upon diagnosis. Instead, it is essential to understand the current state of the entire cardiovascular system before making treatment decisions, which may not always involve the aortic valve itself. Perhaps the authors can consider incorporating these clinical practicalities into their synthesis and writing.
4. The SAVR and TAVR columns in Table 1 appear to be misaligned. I would recommend reviewing and correcting the alignment before submitting the article.
Author Response
Reviewer 1.
This is a review article where the authors aim to elucidate the diagnostic challenges surrounding aortic stenosis. They discuss the various possibilities that need to be considered and provide commentary on different imaging examinations, sharing their professional opinions. The topic is significant, and the English language used is fluent. Below, I will provide my considerations regarding the overall structure and specific details.
- The authors used the term "dilemma" in the title, but the actual meaning of dilemma is a situation where there are two difficult choices, both with pros and cons, making it extremely challenging to choose between them. The main difficulty lies in the fact that "neither option is ideal" and one has to be chosen. However, after reading the abstract and content, it seems that the authors intended to convey the meaning of "challenge" rather than the concept of "either-or" dilemma. Would you consider reconsidering the title?
If the authors' use of "dilemma" is intended to refer to choosing between SAVR or TAVR, it doesn't seem to align with the context. This is because not all cases of aortic stenosis necessarily require treatment, and even when treatment is necessary, it doesn't always involve these two options.
We thank this Reviewer for this comment. We agree with your perspective that the term "dilemma" may imply a binary choice, which could potentially mislead our readers regarding the intent of our paper. Our intention was to represent the complexity and difficult nature of evaluating and treating patients with aortic stenosis who also present with multivalvular disease. In light of your comments, we propose a revised title:
“Beyond Aortic Stenosis: Addressing the Challenges of Multivalvular Disease Assessment”
This revised title better reflects the comprehensive review and discussion we aim to provide in our paper, emphasizing the challenge rather than a strict either-or situation.
- In terms of overall structure, the authors have extensively explored and expanded upon the topic of aortic stenosis. This is indeed reflective of the clinical reality, as aortic stenosis is often not an isolated condition. Even when considering the management of aortic stenosis alone, it is crucial to first exclude other potential underlying causes. Failing to do so may result in worsening of the patient's condition or even increased mortality, highlighting the importance of a comprehensive approach.
However, the current structure of the article seems somewhat divergent. The authors have indeed presented various diseases that are potentially related to aortic stenosis, some being sequelae following aortic stenosis and others being the causes of aortic stenosis. In the end, the authors conclude that each patient's condition should be evaluated individually. However, this organization lacks clear recommendations or standardized guidelines to follow. It appears to primarily provide a wealth of foundational knowledge on cardiac diseases and advises readers to exercise their own judgment.
I would suggest that the authors consider reorganizing the article by categorizing the important scenarios they believe are valuable for readers. Each scenario can be introduced with a case study, particularly emphasizing those cases where misdiagnosis could lead to poor outcomes or those that are often overlooked. This approach would not only showcase your team's expertise in the field but also provide recommendations for addressing the challenging and unstructured clinical dilemmas in a more focused manner.
We thank the Reviewer for your detailed feedback and for the suggestions to enhance our paper's structure. We appreciate the proposed idea of incorporating case studies to create a more focused narrative. However, we believe that including specific cases may complicate the text, potentially confusing the reader, and significantly lengthen the manuscript. Nonetheless, we agree with the feedback regarding the need for more structured guidance. In response, we have divided the "Treatment" paragraph into two sections. The second part will provide detailed strategies for identifying the predominant valvular lesion in situations where aortic stenosis coexists with other valve diseases. Additionally, we have focused on the most commonly encountered clinical scenarios, namely aortic stenosis with concomitant mitral regurgitation or tricuspid regurgitation, and have proposed algorithms to guide the reader in the therapeutic management of these patients.
- In practical terms, when we encounter patients with aortic stenosis in clinical practice, the usual approach is to perform a comprehensive cardiac ultrasound examination to assess all four cardiac chambers and valve conditions. This evaluation helps determine whether the aortic stenosis is primary or secondary, and provides an understanding of the overall cardiac function. Treatment considerations are then made based on these findings. I believe your team follows a similar approach. In other words, aortic stenosis is not a condition that requires immediate standalone treatment upon diagnosis. Instead, it is essential to understand the current state of the entire cardiovascular system before making treatment decisions, which may not always involve the aortic valve itself. Perhaps the authors can consider incorporating these clinical practicalities into their synthesis and writing.
We thank this Reviewer for the insightful feedback and for sharing his/her valuable clinical perspective. We agree with the practical approach to managing aortic stenosis, emphasizing the importance of a comprehensive cardiac assessment to understand the state of the entire cardiovascular system. This is indeed the perspective we sought to convey through our manuscript. In response to your suggestion, we have integrated these clinical realities into our discussion (lines 430-434). We have explicitly emphasized that treatment decisions must be grounded in a thorough understanding of the overall cardiovascular system. We have underscored the fact that addressing aortic stenosis may not always solely involve interventions on the aortic valve.
- 4. The SAVR and TAVR columns in Table 1 appear to be misaligned. I would recommend reviewing and correcting the alignment before submitting the article.
We thank this Reviewer for bringing this to our attention. We have now ensured the columns for SAVR and TAVR in Table 1 are correctly aligned.
Reviewer 2 Report
Bombace et al. present a review about the pathophysiology of Aortic Stenosis in the context of multiple and mixed valve disease, with a focus on the hemodynamic implications, diagnostic challenges, and therapeutic options.
Although the topic is interesting and the manuscript well-written, some considerations need to be clarified.
1) Abstract: The use of multimodality imaging represents the future in the evaluation of mixed valve disorders, this point should be emphasized.
2) 3.1 Role of echocardiography (Mitral Regurgitation): A high-velocity MR jet may be mistaken for the AS jet as both are systolic signals directed away from the apex. This issue should be described and added in the text, and how the timing of the signal and the shape of the velocity curve can be helpful in differential diagnosis.
3) 3.1 Role of echocardiography (Aortic regurgitation): Please, thoroughly describe the fundamental assessment of anatomic AVA and the use of multimodality imaging in this specific case.
Please, add the following references to improve the quality of the paragraph:
- Marwick TH, et al. JACC Cardiovasc Imaging. 2022 May;15(5):957-959. doi: 10.1016/j.jcmg.2022.04.
- Siani A, et al. J Clin Ultrasound. 2022 Oct;50(8):1041-1050. doi: 10.1002/jcu.23299.
4) 3.1. Role of echocardiography (Tricuspid Regurgitation): In the physiopathology of the AS, functional TR is present in one third of the patients and careful echocardiographic evaluation is required. Please, add the following references to improve the quality of the paragraph:
- Vieitez JM, et al. Eur Heart J Cardiovasc Imaging. 2021 Jan 22;22(2):196-202. doi: 10.1093/ehjci/jeaa205.
- Zoghbi WA, et al. J Am Soc Echocardiogr. 2017 Apr;30(4):303-371. doi: 10.1016/j.echo.2017.01.007
5) Please, add a final paragraph on “future aspects and gaps in evidence” highlighting the future role of multimodality imaging.
Author Response
Reviewer 2.
Bombace et al. present a review about the pathophysiology of Aortic Stenosis in the context of multiple and mixed valve disease, with a focus on the hemodynamic implications, diagnostic challenges, and therapeutic options.
Although the topic is interesting and the manuscript well-written, some considerations need to be clarified.
1) Abstract: The use of multimodality imaging represents the future in the evaluation of mixed valve disorders, this point should be emphasized.
We thank this Reviewer for this comment. We fully agree that the use of multimodality imaging is crucial and will play an increasingly important role in the evaluation of mixed valve disorders. We have emphasized this point in the revised abstract (lines 29-31).
2) 3.1 Role of echocardiography (Mitral Regurgitation): A high-velocity MR jet may be mistaken for the AS jet as both are systolic signals directed away from the apex. This issue should be described and added in the text, and how the timing of the signal and the shape of the velocity curve can be helpful in differential diagnosis.
We thank this Reviewer for this valuable feedback. We agree that addressing this point is crucial, as distinguishing between the two is pivotal for accurate diagnosis. We have incorporated your suggestion and provided further details on differentiating between these signals in our manuscript (lines 231-236).
3) 3.1 Role of echocardiography (Aortic regurgitation): Please, thoroughly describe the fundamental assessment of anatomic AVA and the use of multimodality imaging in this specific case.
Please, add the following references to improve the quality of the paragraph:
- Marwick TH, et al. JACC Cardiovasc Imaging. 2022 May;15(5):957-959. doi: 10.1016/j.jcmg.2022.04.
- Siani A, et al. J Clin Ultrasound. 2022 Oct;50(8):1041-1050. doi: 10.1002/jcu.23299.
We thank this Reviewer for these valuable suggestions. We concur that the assessment of anatomic AVA deserves an in-depth exploration in our discussion. In response to his/her advice, we have elaborated on the role of transesophageal echocardiography in evaluating anatomic AVA (lines 280-286). Descriptions of the use of CT and CMR are already included in the manuscript (lines 358-364 and 385-408, respectively). Moreover, we appreciate the recommended references and have integrated them into our manuscript to augment the depth of our discussion.
4) 3.1. Role of echocardiography (Tricuspid Regurgitation): In the physiopathology of the AS, functional TR is present in one third of the patients and careful echocardiographic evaluation is required. Please, add the following references to improve the quality of the paragraph:
- Vieitez JM, et al. Eur Heart J Cardiovasc Imaging. 2021 Jan 22;22(2):196-202. doi: 10.1093/ehjci/jeaa205.
- Zoghbi WA, et al. J Am Soc Echocardiogr. 2017 Apr;30(4):303-371. doi: 10.1016/j.echo.2017.01.007
We thank this Reviewer for this insightful comment. We appreciate the suggested references and have integrated these into the corresponding section of the manuscript to further enhance its accuracy and quality.
5) Please, add a final paragraph on “future aspects and gaps in evidence” highlighting the future role of multimodality imaging.
We thank this Reviewer for this important suggestion. We have made an improvement by adding a paragraph at the end to address future aspects and gaps in evidence, specifically to acknowledge the potential role of multimodality imaging.
Round 2
Reviewer 1 Report
The authors have responded to the suggestions regarding the title and the previous recommendations on structure and formatting details. While there haven't been significant modifications in terms of the overall structure, they have made minor adjustments to address the various suggestions put forward earlier, which has been done quite well.
I have no further questions.
Fluent
Reviewer 2 Report
The authors answered my questions perfectly.
Congratulations on a really interesting and impactful manuscript.